# Tumor-Associated Macrophages in Hematologic Malignancies: New Insights and Targeted Therapies

**DOI:** 10.3390/cells8121526

**Published:** 2019-11-27

**Authors:** Amy J. Petty, Yiping Yang

**Affiliations:** 1Department of Pharmacology and Cancer Biology, Duke University, Durham, NC 27710, USA; amy.petty@duke.edu; 2Division of Hematology, The Ohio State University Wexner Medical Center, 508 BRT, 460 W 12th Avenue, Columbus, OH 43210, OH, USA

**Keywords:** tumor-associated macrophage, nurse-like cells, leukemia, lymphoma, myeloma, immunotherapy

## Abstract

The growth of hematologic malignant cells can be facilitated by other non-tumor cells within the same microenvironment, including stromal, vascular, immune and mesenchymal stem cells. Macrophages are an integral part of the human innate immune system and the tumor microenvironment. Complex interplays between the malignant hematologic cells and the infiltrating macrophages promote the formation of leukemia, lymphoma or myeloma-associated macrophages. These pro-tumorigenic macrophages in turn play an important part in facilitating tumor growth, metastasis and chemotherapeutic resistance. Previous reports have highlighted the association between tumor-associated macrophages (TAMs) and disease progression in hematologic malignancies. This review summarizes the role of TAMs in different subtypes of leukemia, lymphoma and myeloma, focusing on new insights and targeted therapies.

## 1. Introduction

In all stages of tumor development, various types of immune cells, including lymphocytes and macrophages coinhabit with malignant cells. The abundance of tumor infiltrating macrophages, termed tumor-associated macrophages (TAMs) in an array of solid tumors have been highlighted in previous reports [1]. As a critical component of the human innate immune system, it was originally thought that the presence of TAMs contributes to tumor surveillance and eradication. However, recent evidence suggests that TAMs are paradoxically involved in tumorigenesis and may contribute to neoplastic progression through a variety of mechanisms, including promoting angiogenesis, metastasis, cancer stemness and local immunosuppression within the tumor microenvironment (TME) [2,3]. Hence, TAM infiltration and its polarization to a pro-tumorigenic and immunosuppressive phenotype are correlated with poor prognosis in many types of human cancer [1].

Increasing evidence also underlined the important role TAMs play in hematologic malignancies [4,5]. Within the TME of leukemia, lymphoma or myeloma, there are dynamic interactions between immune cells, including TAMs and the malignant hematologic cells. Under the influence of tumor cells, these TAMs are then reprogrammed into an immunosuppressive and pro-tumorigenic phenotype, which can in turn subvert immune responses and contribute to accelerated tumorigenesis. This review focuses on recent discoveries on the role of TAMs in hematologic malignancies.

## 2. Macrophages and Tumor-Associated Macrophages

Macrophages are cells of the hematopoietic system that participate in many important functions in the body, ranging from inflammation, tissue repair to homeostasis [6]. As macrophages are highly responsive to their environmental stimuli, there is a great level of heterogeneity in macrophage morphology, phenotype and function. Activated macrophages have been divided into two broad categories—the classically activated M1 macrophages and the alternatively activated M2 macrophages [7]. Though these states mirror the T helper 1 (Th1)-Th2 states of T cells, the M1/M2 phenotypes are not stably differentiated states in the same way as Th1 and Th2 cells. Rather, macrophages can exhibit phenotypes in a spectrum of possible forms with two extremes defined by M1 and M2 [8]. M1 macrophages, activated by lipopolysaccharide (LPS), interferon-γ (IFN-γ), granulocyte-macrophage colony stimulating factor (GM-CSF), preferentially secrete pro-inflammatory molecules, such as IL-1, IL-6, IL-12, tumor necrosis factor-α (TNF-α), CXCL9, CXCL10, nitric oxide and reactive oxygen species. Due to their high capacity to antigen present and elicit a strong Th1 response, M1 macrophages are considered antitumoral. In contrast, M2 macrophages are activated by IL-4/IL-13, IL-10 and macrophage colony stimulating factor (M-CSF; also known as CSF1) and in turn express high levels of anti-inflammatory molecules, such as IL-10, transforming growth factor-β (TGF-β), CCL17, CCL22, arginase, mannose (CD206) and scavenger (CD163) receptors. As such, M2 macrophages are thought to play an immunosuppressive role in the body [7,8,9].

The formation of pro-tumorigenic TAMs in the tumor stroma requires—1) factors that facilitate monocyte/macrophage recruitment and expansion; and 2) factors that drive the TAM polarization into the immunosuppressive phenotype [10]. Macrophage infiltration into the tumor stroma is mediated by chemokines CCL2, CCL5, CCL7 and CX3CL1 as well as molecules such as M-CSF and vascular endothelial growth factor (VEGF) [11,12,13,14]. Once recruited to the tumor site, various tumor-derived or stroma-derived factors orchestrate the immunosuppressive and pro-tumorigenic polarization of TAMs [15,16]. IL-4 and IL-13 act on TAMs through intracellular signal transducer and activator of transcription 6 (Stat6) and PI3K signaling while IL-10 produced by regulatory T cells (T_reg_) also promotes the TAM phenotype via the actions of Stat3 [17]. Other molecules secreted by the tumor cells, including TGF-β, lactic acid and sonic hedgehog (Shh) also strongly promote M2-polarization of TAMs, resulting in subsequent promotion of tumorigenesis [18,19,20]. It is important to note that TAMs are a group of highly heterogeneous cells that have diverse functions and phenotypes depending on cancer type and tumor stage. In addition to classifying TAMs based on previously described M2-macrophage markers, recent attempts to categorize TAMs based on their functional role in the tumor microenvironment have also gained momentum [1]. Therefore, the M1–M2 dichotomy is excessively simplistic for TAM classification and additional studies are needed to better clarify TAM classification and subgroups within the TME of different cancer types.

The presence of macrophages in the TME of leukemia, lymphoma or myeloma can be suggestive of several signs, including prognosis of disease progression and efficacy of chemotherapy. Once in the TME, TAMs can interact with other cells through different mechanisms to create a pro-tumorigenic environment—1) stimulate tumor cell growth and metastasis directly by producing growth and matrix remodeling factors; 2) promote chemoresistance in cancer cells and 3) induce immunosuppression by altering the behavior of innate or adaptive immune cells [21]. Due to its critical role in the TME, its role in solid tumors has been extensively investigated. However, due to the unique and diverse microenvironments involved in hematologic malignancies, the importance of TAMs only began to emerge. We will focus on recent preclinical and clinical discoveries.

## 3. The Role of TAMs in Leukemia

Recent publications have emphasized the role of TAMs in three subtypes of leukemia, including acute lymphoblastic leukemia (ALL), acute myeloid leukemia (AML) and chronic lymphocytic leukemia (CLL).

### 3.1. Acute Lymphoblastic Leukemia

Evaluating the phenotype of CD163^+^ M2-like TAMs in 58 acute T-lymphoblastic leukemia (T-ALL) patients revealed that higher numbers of CD163^+^ cells were correlated with poor prognosis. In addition, the percentage of CD163^+^ cells was an independent prognostic factor in these patients. Co-culturing M2-macrophages with T-ALL cell in vitro significantly induced leukemic cell proliferation via C5a, TNFα, growth-related oncogene (GRO)-α and IL-6 [22]. Hohtari et al. analyzed immune cell constitution in adult precursor B cell ALL bone marrow (BM), demonstrating increased proportion of M2-like macrophages and myeloid-derived suppressor cells (MDSCs) in ALL patients’ BM compared to healthy patients [23].

Further analysis of different lymphoid organs in a Notch1-overexpressing T-ALL mouse model revealed different phenotypes and gene expression patterns of TAMs in BM vs. spleen. Though both capable of expressing M2-associated genes, splenic TAMs stimulated the proliferation of T-ALL cells more potently compared to BM-TAMs, demonstrating the highly plastic nature of TAMs under different microenvironments [24]. Valencia et al. also reported that ALL cells released Bone Morphogenetic Protein 4 (BMP4) to induce immunosuppressive dendritic cells and generate M2-like macrophages, which produced low TNFα and high CCL2, IL-6 and IL-10 [25]. Lastly, a recent report found that stromal interaction molecule 1 (STIM1) and STIM2 mediated the pathologic cancer-induced inflammation in the TME of T-cell ALL. Deletion *Stim1* and *Stim2* in malignant ALL cells not only reduced the number of infiltrating macrophages but also rescued the pro-inflammatory phenotype of TAMs through IFNγ [26]. Together, these reports highlighted that interactions between leukemic cells and TAMs may be important in promoting tumorigenesis in ALL.

### 3.2. Acute Myeloid Leukemia

Al-Matary et al. evaluated the role of TAMs in AML tumorigenesis and found that the frequency of CD163^+^CD206^+^ M2-like TAMs was significantly elevated in the BM of AML patients compared to healthy volunteers. Using different murine models of AML, they also found that leukemic cells polarized TAMs to an M2-like phenotype, which subsequently accumulated in the BM and spleen of tumor-bearing mice. Conversely, bone marrow-derived macrophages (BMDMs) from leukemic mice supported the in vitro expansion of AML cells better than that from non-leukemic mice. They also discovered the critical role of growth factor independent 1 (Gfi1) transcriptional repressor in polarizing TAMs toward a pro-tumorigenic M2-state in vitro and in vivo [27]. Yang et al. further confirmed that the number of CD163^+^ M2-like TAMs was correlated with worse prognosis in AML patients with splenic TAMs exhibiting more M2-characteristics than BM-TAMs. Additionally, they found that Interferon Regulatory Factor 7 (IRF7) contributed to the M1-polarization of TAMs through activation of the SAPK/JNK pathway and subsequent activation of the IRF7-SAPK/JNK pathway resulted in more M1-like TAMs, which was correlated with prolonged survival in leukemic mice [28].

Lastly, a recent report by Jiang et al. highlighted the role of monocytic leukemia zinc-finger protein (MOZ) in the differentiation and M1-polarization of macrophages in AML. A low level of MOZ was associated with poor prognosis in AML patients and genetic silencing of MOZ suppressed M1 activation of macrophages. Furthermore, miR-223, a microRNA that was previously shown to suppress M1-polarization and play an important role in the pathogenesis of AML can regulate MOZ functions [29]. Collectively, these reports provided some evidence for the importance role M2-like TAMs play in the progression of AML.

### 3.3. Chronic Lymphocytic Leukemia

The role macrophages play in CLL was first discovered in 2000 when Burger et al. found that differentiated peripheral mononuclear cells from B-cell CLL patients could protect CLL cells from undergoing spontaneous apoptosis through the action of stromal cell-derived factor-1 (SDF-1; also known as CXCL12) in vitro and the authors coined the term “nurse-like cells” (NLCs) [30]. CXCL13 can also be released by the CD68^+^ NLCs to support CLL migration and growth through the activation of p44/42 mitogen-activated protein kinases (MAPKs) downstream of CXCR5 [31]. It was later discovered that NLCs were a critical component of the leukemic microenvironment in CLL and phenotypically and functionally equivalent to TAMs in solid tumors with high expressions of CD11b, CD68 and CD163 [32,33]. Additionally, under the influence of the hepatocyte growth factor (HGF) released by leukemic cells, c-Met^+^ NLCs exhibited the immunosuppressive functions of M2-like TAMs by inhibiting T-cell proliferation through the action of TGF-β, IL-10 and indoleamine 2,3-idoxygenase (IDO) and supporting Foxp3^+^ T regulatory (T_reg_) cell expansion [34]. Using the Eμ-TCL1 mouse model of CLL, Hanna et al. found that macrophages accumulated in the peritoneal cavity and spleen of leukemic mice in a CCR2-dependent manner and exhibited the M2-like phenotype with a high expression of Programmed Death Ligand-1 (PD-L1). Depletion of myeloid cells in CLL mice using liposomal Clodronate resulted in reduced tumorigenesis and repaired the activation of T cells, demonstrating the extensive immunosuppressive functions of M2-like TAMs in CLL [35].

Examination of cross-talks between the leukemic cells and TAMs revealed that CLL cells could release nicotinamide phosphoribosyltransferase (NAMPT) to induce the M2-phenotype in TAMs through the actions of Stat3 and NF-κB signaling. These CD163^hi^CD206^hi^ macrophages expressed IDO, IL-10. CCL18, IL-6 and IL-8 to support leukemic growth and suppress effector cell responses [36]. Galletti et al. also found that leukemic cells induced the M2-polarization of TAMs in CLL through the colony-stimulating factor 1 (CSF1)-CSF1R pathway and targeting of macrophages by CSF1R blockade reduced leukemic cell load in the BM and prolonged survival [37]. Spontaneously dying CLL cells may also release high-mobility group box 1 (HMGB1) protein to promote the differentiation of monocytes into NLCs/TAMs. Coincidently, the presence of HMGB1 protein and DNA in patient plasma was positively correlated with tumor burden and adverse clinical outcome [38]. Lastly, CLL cells were found to release CCL3 and CCL4 as a result of B-cell receptor (BCR) stimulation by TAMs and the two chemokines worked to recruit other leukemic cells or macrophages to the tumor site [39]. Together, these reports suggested that targeting TAMs in CLL might be therapeutically efficacious. Indeed, simply reprogramming TAMs toward an M1-phenotype with IFN-γ enhanced FcγR-mediated production of cytokines and rituximab-mediated phagocytosis of CLL cells in vitro [40]. Eliminating both offending parties by co-targeting of TAMs and CD20^+^ leukemic cells with blocking antibodies helped reprogram the tumor microenvironment and conferred a synergistic survival benefit in the mouse model [37].

Collectively, this further establishes the important role of macrophages in the tumorigenesis of leukemia and suggests that therapies targeting the leukemia-macrophage interactions could be highly effective clinically. However, further investigation is needed to investigate the diverse molecular mechanisms of TAMs in distinct microenvironments of different types of leukemia.

## 4. The Role of TAMs in Lymphoma

Similar to the role of TAMs in leukemia, macrophages are a critical inducer of lymphoma progression, both in classic Hodgkin’s lymphoma (CHL) and non-Hodgkin’s lymphoma (NHL).

### 4.1. Classic Hodgkin’s Lymphoma

The TME of CHL is composed of mainly inflammatory immune cells and only a small percentage of the cells, approximately 1% are malignant Hodgkin Reed-Sternberg (HRS) cells [41]. Thus, the role played by macrophages is especially important in the pathogenesis of CHL. A study by Steidl et al. demonstrated that increased number of CD68^+^ TAMs in lymph nodes was strongly correlated with decreased progression-free survival (PFS) and more frequent relapse after autologous hematopoietic cell transplantation in CHL patients [42]. Evaluation of CD163, an M2-marker in patient samples of advanced CHL showed that CD68^+^CD163^+^ TAMs were correlated with inferior overall survival [43]. In addition, the percentage of CSF1R^+^ TAM was also inversely associated with survival in CHL patients [44]. A meta-analysis by Guo et al. further confirmed that a higher density of CD68^+^ or CD163^+^ TAMs was a strong predictor of clinical outcome in patients with CHL [45].

In addition, the majority of PD-L1 expression in the TME of CHL was contributed by TAMs. These PD-L1^+^ macrophages colocalized with PD-L1^+^ HRS cells and were in close contact with PD-1^+^ T cells [46]. Lastly, Vari et al. showed that PD-L1/PD-L2^+^ TAMs could suppress activation of PD-1^hi^ natural killer (NK) cells, a process that can be reversed by PD-1 blockade. As a result, depletion of circulating monocytes from the blood of pretherapy patients with CHL enhanced CD3^−^CD56^hi^CD16^−^ NK activation, suggesting a critical role for TAMs in immune evasion and subsequently progression of CHL [47]. Though mechanistic evidence focused on the signaling events within TAMs is lacking, new reports have highlighted the effects of PI3K inhibition on macrophage M2-polarization and metabolic switch in CHL, suggesting the PI3K-Akt pathway could play an important role in TAMs within the TME of Hodgkin’s lymphoma [48]. More research is needed to investigate how CHL cells and the surrounding environment drive the immunosuppressive M2-polarization of TAMs.

### 4.2. Non-Hodgkin’s Lymphoma

The prognostic value of TAMs in follicular lymphoma (FL) was demonstrated in two early reports, which showed that increased CD68^+^ TAM count was associated with poor survival in FL patients [49,50]. This was proposed to be associated with M2 polarization of macrophages. Indeed, literatures also showed that increased microvascular density and angiogenic sprouting were positively correlated with increased CD163^+^ TAMs and poor outcome in FL [51]. This polarization of TAMs can be directly induced by apoptotic lymphoma cells and M2-TAMs expressed decreased level of galectin-3, a pleiotropic glycoprotein involved in apoptotic cell clearance, ultimately resulting in more aggressive progression of NHL [52]. Additionally, mesenchymal stromal cells (MSCs) in the BM of FL overexpressing CCL2 can also participate in driving macrophage toward a pro-angiogenic and immunosuppressive M2-phenotype [53]. In turn, these immunosuppressive macrophages can trans-present IL-15 through a T cell-derived CD40L-dependent mechanism to induce proliferation of follicular lymphoma cells [54].

In human diffuse large B cell lymphoma (DLBCL), though the first-line chemotherapeutic regimen including cyclophosphamide, doxorubicin, vincristine and prednisone (CHOP) is fairly successful in terms of remission rate, it is often followed by a relapse rate of 40% within 2–3 years [21]. As inflammatory response gene signatures in the TME are defining features of DLBCL, studies also found a strong correlation between these inflammatory signals and poor prognosis or treatment resistance in patients with DLBCL [55,56]. It was found that one of two TME signatures, stromal-2 contained signature genes encoded for well-known markers of TAMs and MDSCs and can predict clinical outcomes in DLBCL patients [55,57,58]. Other reports confirmed the prognostic value of TAMs by showing that increased TAMs or M2-TAMs was related to worse prognosis and poorer clinical outcomes in DLBCL and DLBCL of the central nervous system [59,60,61,62]. In addition, the presence of CD204^+^ macrophages was associated with a higher relapse rate and poorer survival in malignant lymphoma patients treated with allogeneic hematopoietic stem cell transplantation [63]. The immunosuppressive CD14^+^HLA-DR^low/−^CD120b^low^ monocytes/macrophages can suppress T cell proliferation through the overexpression of Arg-1, a marker for M2 TAMs and an enzyme that depletes essential nutrients for T lymphocytes. In addition, these macrophages had decreased Stat1 phosphorylation, a transcription factor that has been proposed to modulate M1-polarization of TAMs [8,64]. The expansion of this population of cells was later shown to be induced by IL-10 [65]. Lastly, M2-TAMs can directly remodel the extracellular matrix in DLBCL through legumain, an asparaginyl endopeptidase that leads to degradation of fibronectin and collagen I and increased angiogenesis. Inhibiting legumain action led to decreased tumor growth in a xenograft DLBCL model with reduced angiogenesis and collagen deposition in the TME [66]. Taken together, these reports indicated the active role TAMs play in the progression of DLBCL.

In summary, these reports highlighted the complex interactions between lymphoma cells and TAMs within the microenvironment. Further analysis is needed to investigate the underlying how lymphoma cells drive the phenotypic and functional shift seen in TAMs to better inform future therapeutic developments.

## 5. The Role of TAMs in Myeloma

TAMs were also found to play a critical role in multiple myeloma (MM), a malignant B-cell cancer involving the uncontrolled growth of plasma cells in the BM. In an early study, Zheng et al. found that CD68^+^ macrophages heavily infiltrated the BM of MM patients compared to healthy controls. These macrophages supported proliferation and suppressed apoptosis of myeloma cells in the presence of chemotherapy by inhibiting activation and cleavage of caspase-3 and poly-ADP ribose polymerase (PARP) and maintaining the levels of Bcl-xL in vitro [67]. In addition, macrophages, working in conjunction with MSCs were found to support the proliferation and survival of MM cells through IL-6 and IL-10 production [68]. An in vivo evidence emerged in 2013 where Zheng et al. showed that TAMs contributed to myeloma drug resistance through P-selectin glycoprotein ligand-1 (PSGL-1)/selectins and intercellular adhesion molecule-1 (ICAM-1)/CD18 contact-dependent interactions. These interactions between infiltrating TAMs and myeloma cells resulted in activation of non-receptor tyrosine kinase Src, Erk1/2 kinases and c-myc activation in MM cells, all of which have been shown to promote MM cell survival and drug resistance [69,70]. Together, these reports provided early evidence for the important role TAMs play in MM tumor progression. The prognostic value of CD68^+^CD163^+^ TAMs in MM was realized when Suyani et al. found that number of TAMs was associated with high-grade micro-vessel density and decreased survival [71]. Two reports further reported a negative correlation between CD163 and CD206 expressions, two M2-macrophage markers and overall survival in patients with MM [72,73]. Recently, high number of CD163^+^ TAMs was also correlated with poor prognosis characterized by decreased PFS and complete remission/near-complete remission in MM patients receiving bortezomib-based chemotherapy [74]. Collectively, there is strong literature evidence demonstrating the close relationship between TAMs and MM. However, there is no report demonstrating the intracellular events occurring in TAMs as a result of these interactions. Potential therapeutic strategies should also be explored to better target macrophages in treatment of MM.

## 6. New Macrophage-Targeting Therapies in Hematologic Malignancies

Many clinical approaches to therapeutically target TAMs are currently under investigation. To prevent immunosuppression and tumor-promotion mediated by mainly M2-macrophages, three general approaches are currently believed to be effective—depletion, reprogramming and molecular targeting. This review will focus on some of the recent developments in macrophage-targeting therapies in hematologic malignancies.

It is well known that CSF1R signaling in macrophages is essential for its recruitment, differentiation and survival. The loss of CSF1-CSF1R signaling can significantly reduce the number of TAMs in mouse models of various solid tumors [10]. Therefore, it was proposed that blockade of CSF1R can reduce macrophage accumulation and decrease its immunosuppressive functions. Indeed, in B-cell CLL, CSF1R inhibition with a JAK2/FLT3 inhibitor, pacritinib was associated with significant depletion of TAMs and consequently inhibited leukemic cell survival [75]. A similar effect was recently observed in AML where CSF1R inhibition reduced paracrine growth signals from the TME, resulting in reduced leukemic cell viability [76]. In MM, blocking CSF1R with antibody or depleting macrophages both led to reduced tumor growth and sensitized myeloma cells to chemotherapy in vivo [77]. The antitumoral effects of CSF1 blockade was also observed in mantle cell lymphoma where treatments with GW2580, a CSF1R inhibitor reduced lymphoma cell survival, irrespective of their sensitivity to ibrutinib, a Bruton’s tyrosine kinase (BTK) inhibitor [78]. Lastly, synergistic effects were observed when combining CSF1R inhibitors with idelalisib or ibrutinib, two current CLL therapies that target PI3K and BTK, respectively [79]. Together, these reports highlighted the therapeutic potential of CSF1R blockade in treatments of leukemia or lymphoma, but further investigation is needed in the clinical setting to better define its role.

Other agents that target the CCL2 pathway include trabectedin and lenalidomide. Trabectedin can selectively kill monocytes and/or macrophages by inhibiting CCL2, CXCL8, IL-6 and VEGF production in vitro [80]. Trabectedin treatment resulted in selective killing of monocytes/macrophages in blood, spleen and tumors while sparing neutrophils and lymphocytes via a caspase-8-dependent apoptosis pathway. This was also associated with downregulation of CCL2 and VEGF, resulting marked decrease in the recruitment of TAMs to the TME and reduced angiogenic sprouting in vivo [81]. While trabectedin is currently approved for the treatment of soft tissue sarcoma and relapsed platinum-sensitive ovarian cancer in the US, its potential role has been proposed in MM [82]. Due to its ability to functionally reshape macrophages and TAMs, its role should be further investigated in other macrophage-rich hematologic malignancies.

The use of lenalidomide, an immunomodulatory agent has been implicated in B-cell malignancies including CLL and MM. Though the exact mechanism of lenalidomide’s antitumor activity is not completely understood, it has been reported to downregulate anti-inflammatory and proangiogenic cytokines and modify the TME via promotion of T and NK cell functions [83]. An early study showed lenalidomide suppressed the secretion CCL2 from NLCs and inhibited the survival support of NLCs for CLL cells in vitro [84]. A recent in vivo study showed that lenalidomide in combination with dendritic cell vaccination and anti-PD-1 blocking antibody decreased M2-TAMs with an associated reduction in TGF-β and IL-10 in the spleen of myeloma-bearing mice [85]. In several clinical trials for B-cell lymphoma, lenalidomide showed promising clinical efficacy either as a single agent or in combination with other immunotherapeutic agents [86,87,88,89]. Together, this provided exciting evidence for the unique immunomodulatory properties in leukemia and lymphoma. Further investigation should focus on elucidating its mechanism of action in the TME and better defining its role in additional clinical settings.

One exciting new approach to target macrophages in leukemic or lymphoid malignancies involves targeting the CD47 molecule, a surface glycoprotein that is commonly expressed on malignant cells. When bound to its receptor signal regulatory protein alpha (SIRPα) on myeloid cells, it inhibits macrophage phagocytosis and allows for evasion of innate immune surveillance [90]. Early studies have demonstrated increased CD47 expression on dysplastic human hematopoietic cells, especially in AML and CLL [91,92]. Preclinical evidence showed that blockade of CD47 promoted macrophage phagocytosis to inhibit growth of myeloma cells and triggered T cell-mediated destruction of tumors through increased dendritic-cell cross-priming [93,94]. In addition, a bispecific antibody with affinity for both CD47 and CD20 recapitulated the synergistic effect of rituximab plus anti-CD47 antibody in vitro and led to significantly prolonged survival in mouse models of NHL [95]. Together, these reports paved the way for exploring the therapeutic potential of targeting CD47 in clinical settings. Several CD47-SIRPα antagonists are currently active in phase I o I/II clinical trials for hematologic malignancies—including Hu5F9-G4, CC-90002, TTI-621 and ALX-148 (Table 1). The first two agents, Hu5F9-G4 and CC-90002 are humanized monoclonal antibodies against CD47 while TTI-621 and ALX-148 are SIRPα-IgGFc fusion proteins that act as decoy receptors [96]. Hu5F9-G4 in combination with rituximab showed promising therapeutic efficacy in a phase 1b trial for patients with DLBCL and FL with 50% of the patients enrolled showed objective complete or partial response [97]. As part of a phase 1a trial, five patients with Sezary syndrome, the leukemic variant of cutaneous T-cell lymphoma were treated with TTI-621. Treatment resulted in significant tumor load reduction and skin improvement [98]. Overall, the use of CD47-targeting therapies in clinical settings has an exciting outlook and more comprehensive analysis of its effects on macrophages and other innate or adaptive immune cells needs to be conducted.

## 7. Conclusion and Perspective

Immunotherapy targeting different innate or adaptive immune cells is extensively studied and many studies have proposed newer ways to stimulate the immune system alone or in combination with pre-existing therapies. Unfortunately, there is no single strategy that works for all tumor types and many differences exist between solid tumors and hematologic malignancies. This is likely related to the unique and diverse tumor microenvironment composition in different leukemia and lymphoma as well as the inherent immunogenicity of different subtypes of hematologic malignancies. It has become apparent now that strategies that focus on enhancing the antitumoral abilities of cytotoxic T lymphocytes (CTLs) is insufficient to achieve significant tumor clearance in leukemia, lymphoma and myeloma. Rather, intervention that targets the immunosuppressive TME must be used in conjunction with CTL activation. As we have summarized in this review, TAMs play a critical role in hematologic malignancies through different mechanisms (Figure 1). Thus, it represents an attractive target in designing immunotherapies and different ways of targeting TAMs in hematologic malignancies have been found to synergistically enhance response to chemotherapeutics or immunotherapeutics. However, elucidating the molecular mechanisms responsible for macrophage polarization, especially in the various diverse TMEs of hematologic malignancies is necessary to determine the most effective TAM-targeting approaches to improve immunotherapies. It would also be valuable to better identify the array of markers and immunosuppressive molecules of TAMs in different leukemia/lymphoma subtypes to better delineate this heterogeneous population of cells. Results from these studies will be clinically informative in the treatment of hematologic malignancies in the future.

## Figures and Tables

**Figure 1 cells-08-01526-f001:**
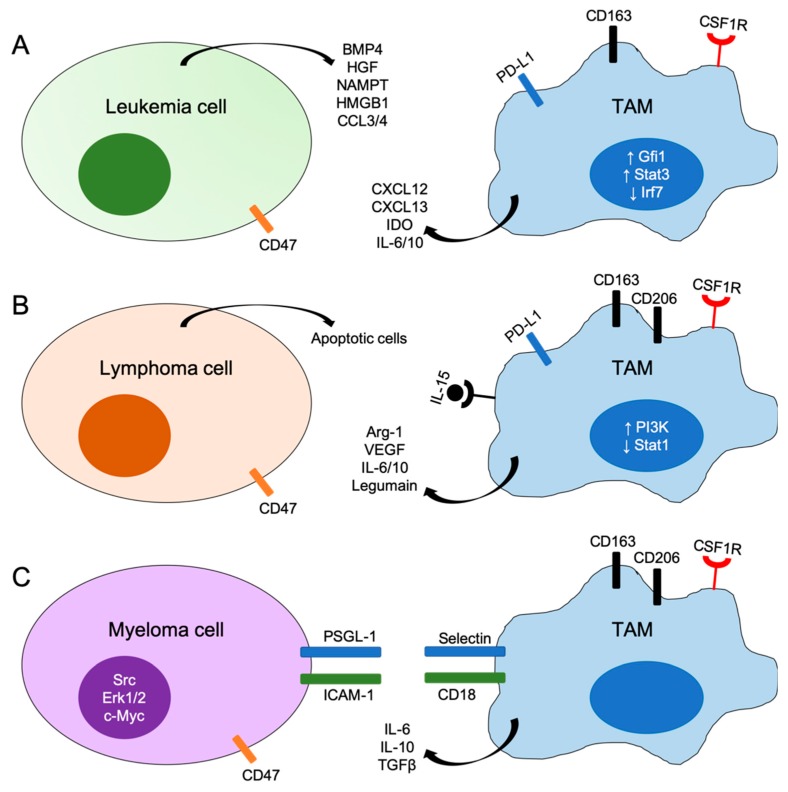
Schematic representations of mechanisms of interactions between tumor cells and tumor-associated macrophages (TAMs) in leukemia (**A**), lymphoma (**B**) and myeloma (**C**). BMP4, bone morphogenetic protein 4; HGF, hepatocyte growth factor; NAMPT, nicotinamide phosphoribosyltransferase; HMGB1, high mobility group box 1; IDO, indoleamine 2,3-dioxygenase; Arg-1, arginase-1; VEGF, vascular endothelial growth factor; PSGL-1, P-selectin glycoprotein ligand-1; ICAM-1, intercellular adhesion molecular-1; Gfi1, growth factor independent 1; Irf7, interferon regulatory factor 7; Stat, signal transducer and activator of transcription; CSF1R, colony stimulating factor 1 receptor.

**Table 1 cells-08-01526-t001:** CD47 antagonists currently in clinical trials for hematologic malignancies.

Agents	Diseases	Open Clinical Trials	Phase	Status and Results
Hu5F9-G4	AML	NCT026783338—CAMELLIA: anti-CD47 antibody therapy in relapsed/refractory AML	I	- Recruitment completed - Hu5F9-G4 infusion in refractory AML patients led to hemoglobin decline and increased transfusion requirements [99].
Hu5F9-G4 + Atezolizumab	AML	NCT03922477—trial of Hu5F9-G4 with atezolizumab in patients with relapsed and/or refractory AML	Ib	- Recruiting
Hu5F9-G4 +/– Azacitidine	AML MDS	NCT03248479—trial of Hu5F9-G4 monotherapy or with azacytidine in hematologic malignancies	I	- Recruiting
Hu5F9-G4 + Rituximab	NHL	NCT02953509—trial of Hu5F9 with rituximab in relapsed/refractory NHL	Ib/II	- Recruiting - Hu5F9-G5 plus rituximab showed promising activity in NHL patients with no clinically significant safety events [97].
Hu5F9-G4 + Rituximab + Acalabrutinib	NHL	NCT03527147—PRISM: platform study for the treatment of relapsed/refractory aggressive NHL	I	- Recruiting
CC-90002	AML MDS	NCT02641002—trial of CC-90002 in patients with AML or high risk MDS	I	- Terminated - Monotherapy did not show encouraging profile for further studies
CC-90002 +/– Rituximab	Solid and hematologic tumors	NCT02367196—trial of CC-90002 in patients with advanced solid tumors and hematologic malignancies	I	- Recruiting
TTI-621 +/– Rituximab or Nivolumab	Solid and hematologic tumors	NCT02663518—trial of TTI-621 alone or with other agents for hematologic malignancies and selected solid tumors	I	- Recruiting - TTI-621 showed therapeutic efficacy for SS patients [98].
ALX-148 +/– Pembrolizumab or Trastuzumab or Rituximab	Solid and hematologic tumors	NCT03013218—trial of ALX-148 in patients with advanced solid tumors and lymphoma	I	- Recruiting

Abbreviations: AML—acute myeloid leukemia; MDS—myelodysplastic syndrome; NHL—Non-Hodgkin’s lymphoma; SS—Sézary syndrome.

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
