# Peer review of "Tumor-Associated Macrophages in Hematologic Malignancies: New Insights and Targeted Therapies"

_cells, 2019, doi:10.3390/cells8121526_

Round 1

Reviewer 1 Report

This is a comprehensive review about tumor-associated macrophages in hematological malignancies. Underlying hypothesis is that M1 macrophages suppress tumor growth and M2 macrophages suppress immune response. Nevertheless, this is a crude classification as authors say. An additional consideration of anti-tumor activities of anti-inflammatory therapies would help understanding this better. The manuscript describes these TAMs in leukemia, lymphoma, and myeloma and ends with therapeutic interventions. A schematic drawing underlines the hypothesis.

Author Response

Reviewer #1:

This is a comprehensive review about tumor-associated macrophages in hematological malignancies. Underlying hypothesis is that M1 macrophages suppress tumor growth and M2 macrophages suppress immune response. Nevertheless, this is a crude classification as authors say. An additional consideration of anti-tumor activities of anti-inflammatory therapies would help understanding this better. The manuscript describes these TAMs in leukemia, lymphoma, and myeloma and ends with therapeutic interventions. A schematic drawing underlines the hypothesis.

Response: Thank you for the suggestion. We found that while anti-inflammatory therapies have proven to be effective in solid tumors, little evidence exists for its role in hematologic malignancies. In addition, we found no report that shows these anti-inflammatory therapies specifically target M1-like TAMs.

Reviewer 2 Report

This is a concise and well organized review on an important topic that has not been frequently discussed in the latter years. There are a few points that would need some modification, listed below.

In the Introduction, description of macrophage activation/polarization is rather simplistic. Since the formulation of the M1 vs. M2 hypothesis, now considered somewhat dogmatic, several refinements of the theory were made and now it is clear that M1 and M2 are the two extremes of a wide continuum. It is mentioned by the authors towards the end of Introduction where they deal with TAM classification but this does not only apply to TAMs but also to macrophages in general, which are also highly heterogeneous. Moreover, in their important and highly cited paper, Xue et al (ref. no. 8 in the manuscript), based on transcriptomic analysis, proposed a new spectrum model of macrophage activation instead of the bipolar M1-M2 one. This should also be mentioned. Besides, ref. 8 is erroneously referred to at description of the M1-M2 theory. The abbreviations LAM and MAM are introduced at the beginning of the manuscript, however, TAM is used almost everywhere in the text for all types of hematologic malignancies. Moreover, in several cases the term TAMs is used even when talking about peripheral blood cells, e.g. in the first sentence of the CLL section; in these cases “TAMs” would be advised to be changed to “macrophages”. There are several papers on the prognostic effect of TAMs in lymphomas, including meta-analyses; these, or some of these should be cited (meta-analyses: Guo B et al BMC Medicine 2016, Xu X et al Scand J Immunol 2019; other recent papers: Li YL et al BMC Cancer 2019, Kawajiri A et al Eur J Haematol 2019, Nam SJ et al Oncoimmunology 2018, Kridel R et al Clin Cancer Res 2015). There are some misspellings and other smaller errors in the text, e.g. cytokines instead of chemokines is written at line 158, and CHL instead of NHL (for non-Hodgkin’s lymphoma) at line 172. Grammatical errors can also be found; a few examples: TAMs followed by singular verbs in lines 27, 28, and 34; “capacity to antigen present” in line 51; “other cells within through” in line 75; missing “that” after “highlighted” (line 106), after “showed” (lines 180 and 196). Language check would be required. Line 254: “there is report…” – the authors probably intended to write “there are no reports”, this would make more sense.

Author Response

Reviewer #2:

This is a concise and well organized review on an important topic that has not been frequently discussed in the latter years. There are a few points that would need some modification, listed below.

In the Introduction, description of macrophage activation/polarization is rather simplistic. Since the formulation of the M1 vs. M2 hypothesis, now considered somewhat dogmatic, several refinements of the theory were made and now it is clear that M1 and M2 are the two extremes of a wide continuum. It is mentioned by the authors towards the end of Introduction where they deal with TAM classification but this does not only apply to TAMs but also to macrophages in general, which are also highly heterogeneous. Moreover, in their important and highly cited paper, Xue et al (ref. no. 8 in the manuscript), based on transcriptomic analysis, proposed a new spectrum model of macrophage activation instead of the bipolar M1-M2 one. This should also be mentioned.

Response: We have further discussed the excessive simplicity of the M1-M2 nomenclature of macrophages/TAMs in our introduction and cited the Xue et al. manuscript (lines 47-51 & 72-74).

Besides, ref. 8 is erroneously referred to at description of the M1-M2 theory.

Response: A manuscript that describes the M1-M2 nomenclature is now correctly cited (ref 7) (line 47).

The abbreviations LAM and MAM are introduced at the beginning of the manuscript, however, TAM is used almost everywhere in the text for all types of hematologic malignancies. Moreover, in several cases the term TAMs is used even when talking about peripheral blood cells, e.g. in the first sentence of the CLL section; in these cases “TAMs” would be advised to be changed to “macrophages”.

Response: The LAMs/MAMs nomenclature was removed, and we used TAMs consistently throughout the paper. We also went through the manuscript and made sure the terms “macrophages” and “TAMs” were used appropriately as the reviewer has suggested.

There are several papers on the prognostic effect of TAMs in lymphomas, including meta-analyses; these, or some of these should be cited (meta-analyses: Guo B et al BMC Medicine 2016, Xu X et al Scand J Immunol 2019; other recent papers: Li YL et al BMC Cancer 2019, Kawajiri A et al Eur J Haematol 2019, Nam SJ et al Oncoimmunology 2018, Kridel R et al Clin Cancer Res 2015).

Response: We have cited recent publications that highlighted the prognostic role of TAMs in lymphoma as the reviewer has suggested: Guo B et al BMC Medicine 2016 (citation #45), Nam SJ et al Oncoimmunology 2018 (citation #61), Li YL et al BMC Cancer 2019 (citation #62), Kawajiri A et al Eur J Haematol 2019 (citation #63).

There are some misspellings and other smaller errors in the text, e.g. cytokines instead of chemokines is written at line 158, and CHL instead of NHL (for non-Hodgkin’s lymphoma) at line 172. Grammatical errors can also be found; a few examples: TAMs followed by singular verbs in lines 27, 28, and 34; “capacity to antigen present” in line 51; “other cells within through” in line 75; missing “that” after “highlighted” (line 106), after “showed” (lines 180 and 196). Language check would be required. Line 254: “there is report…” – the authors probably intended to write “there are no reports”, this would make more sense. 

Response: We have fixed all misspellings and grammatical errors in the text.